# Augment your batch: better training with larger batches

## Abstract

Recently, there is regained interest in large batch training of neural networks, both in theory and practice. New insights and methods allowed certain models to be trained using large batches with no adverse impact on performance. Most works focused on accelerating wall clock training time by modifying the learning rate schedule, without introducing accuracy degradation.

We propose to use large batch training to boost accuracy and accelerate convergence by combining it with data augmentation. Our method, "batch augmentation", suggests using multiple instances of each sample at the same large batch. We show empirically that this simple yet effective method improves convergence and final generalization accuracy. We further suggest possible reasons for its success.

## 1 Introduction

### 1.1 Large batch training of neural networks

Neural network training is known to be highly compute intensive, often requiring large amounts of resources, preferably utilizing as much parallelism as possible. Training is usually done using stochastic-gradient-descent (SGD) or one of its variants, where the gradient for each sample is computed separately and accumulated for each weight update. Using this property, the most straightforward way to scale training is by parallel computation of gradients over many samples, also known as "data parallelism" (Krizhevsky, 2014).

While data parallelism by batch computation is common, very large batches ($> 1000$) were avoided by practitioners, as they were claimed to cause an inherent generalization issue, later described as a tendency to reach "sharp minima" (Keskar et al., 2017). This so-called "generalization gap" was lately challenged by Hoffer et al. (2017) which identified it to be mostly caused by the reduction in optimization updates in large batch training rather than some inherent property.

Recent approaches by Hoffer et al. (2017), Goyal et al. (2017), You et al. (2017) and others showed that equally good and even better generalization could be reached in some cases using large batches, by adapting the optimization regime used. These methods often use higher learning rates than previously used, to account for the lower gradient variance in large batch updates. Hoffer et al. (2017) argued that the quality of the optimized model stems from the number of SGD iterations, rather than the number of cycles through the training data (epochs). Naturally, using large batches reduces the number of iterations in each epoch, and so additional epochs may be needed to reach the same quality of a model trained with a smaller batch.

Ott et al. (2018) demonstrated that large batch training could be beneficial even when batch size does not fit into the device and its update is performed by multiple accumulations. Using this method, they report added stability due to reduced gradient noise, as well as lower communication overhead.

In contrast, Masters & Luschi (2018) suggested that small batch updates may still provide benefits over large batch ones, showing better results over several tasks, with higher robustness to hyper-parameter selection such as the learning rate used.

With more available computation power, larger batch sizes can allow shorter training time. So far, this motivation is what primarily made large batch training appealing. Here, instead, we propose a

novel way of utilizing large batch such that the use of large batch improves generalization performance of the final model.

As modern hardware scales up by increasing the computational resources rather than the bandwidth available, small batch updates are causing a noticeable under-utilization of these devices. Because of that, for a given hardware specification it is possible to train with larger batch sizes without a significant time penalty (You et al., 2017; Grave et al., 2016).

With this notion, we show that it is possible to improve model accuracy by utilizing available resources to the maximum using larger batch sizes.

## 1.2 ROLE AND SIGNIFICANCE OF DATA AUGMENTATIONS

A common practice in training modern neural network is to use data augmentations – multiple instances of input samples, each with a different transformation applied to it. For example, on image classification tasks, for any input image, a random crop of varying size and scale is applied to it, together with potentially rotation, mirroring and even color jittering (Krizhevsky et al., 2012). Data augmentations were repeatedly found to provide efficient and useful regularization, often accounting for significant portion of the final generalization performance (Zagoruyko, 2016; DeVries & Taylor, 2017).

Several works even tried to learn how to generate good data augmentations. For example, Bayesian approaches based on the training set distribution (Tran et al., 2017), generative approaches based on generative adversarial networks (Antoniou et al., 2017; Sixt et al., 2018) and search methods aimed to find the best data augmentation policy (Cubuk et al., 2018). Our approach is orthogonal to those methods; thus we believe we can expect even better results by combining them.

Other regularization methods, such as Dropout (Srivastava et al., 2014) or ZoneOut (Krueger et al., 2016), although not explicitly considered as data augmentation techniques, can be considered as such by viewing them as random transforms over inputs for intermediate layers. These methods were also shown to benefit models in various tasks.

## 2 BATCH AUGMENTATION

In this work, we suggest leveraging the merits of data augmentation together with large batch training, by using multiple instances of a sample in the same batch.

We consider a model with a loss function $\ell(\mathbf{w}, \mathbf{x}_n, \mathbf{y}_n)$ where $\{\mathbf{x}_n, \mathbf{y}_n\}_{n=1}^N$ is a dataset of $N$ data sample-target pairs. $T(\mathbf{x})$ is some data augmentation transform applied to each example – e.g, a random crop from an image.

The common training procedure for each batch consists of the following update rule (here using vanilla SGD with a learning-rate $\eta$ and mini-batch size of $B$, for simplicity):

$$\mathbf{w}_{t+1} = \mathbf{w}_t - \eta \frac{1}{B} \sum_{n \in \mathcal{B}(k(t))} \nabla_{\mathbf{w}} \ell\left(\mathbf{w}_t, T(\mathbf{x}_n), \mathbf{y}_n\right)$$

where $k(t)$ is sampled from $\{1, \ldots, N/B\}$ and we assume for simplicity that $B$ divides $N$.

We suggest to introduce $M$ multiple instances of the same input sample by applying the transform $T_i$, here denoted by subscript $i \in \{1..M\}$ to highlight the fact that they are different from one another.

We now use the slightly modified learning rule:

$$\mathbf{w}_{t+1} = \mathbf{w}_t - \eta \frac{1}{M \cdot B} \sum_{i=1}^M \sum_{n \in \mathcal{B}(k(t))} \nabla_{\mathbf{w}} \ell\left(\mathbf{w}_t, T_i(\mathbf{x}_n), \mathbf{y}_n\right)$$

effectively using a larger $M \cdot B$ batch at each steps, that is composed of $B$ samples augmented with $M$ different transforms each.

We note that this updated rule can be computed either by evaluating on the whole $M \cdot B$ batch or by accumulating $M$ instances of the original gradient computation. Using large batch updates as part of batch augmentations makes no change to the number of SGD iterations that are performed for each epoch.

Batch augmentation (BA) can also be used as transforms over inputs of intermediate layers. For example, we can use the common Dropout regularization method (Srivastava et al., 2014) to generate multiple instances of the same sample in a given layer, each with its dropout mask.

Batch augmentation can be easily implemented in any framework with a reference Py-Torch (Paszke et al., 2017) implementation to be available soon at `https://github.com/paper-submissions/augment-batch`.
To further highlight the ease of incorporating these ideas, we note that BA can be added to any training code by merely modifying the input pipeline – augmenting each batch that is fed to the model.

## 3 EXPERIMENTS

To evaluate the impact of batch augmentation, we used several common datasets and neural network based models. For each one of the models, unless explicitly stated, we tested our approach using the original training regime and data augmentation described by its authors. For simplicity, we did not change the learning rate used, although this is possibly sub-optimal.

### 3.1 CIFAR10/100

The Cifar10 dataset introduced by Krizhevsky (2009) is a popular image classification dataset containing $50,000$ training images, together with a $10,000$ test set. Each image is of size $32 \times 32$ and belongs to one of 10 classes of vehicles and animals. The Cifar100 dataset consists of the same number of training and validation images and the same spatial size, but with an increase to $100$ in the number of possible classes for each image.

For both datasets, we used the common data augmentation technique as described by He et al. (2016). In this method, the input image is padded with $4$ zero-valued pixels at each side, top, and bottom. A random $32 \times 32$ part of the padded image is then cropped and with a $0.5$ probability flipped horizontally. This augmentation method has a rather small space of possible transforms ($9 \cdot 9 \cdot 2 = 162$), and so it is quickly exhausted by even a $M \approx 10$s of simultaneous instances.

We therefore speculated that using a more aggressive augmentation technique, with larger option space, will yield more noticeable difference when batch augmentation is used. We chose to use the recently introduced "Cutout" (DeVries & Taylor, 2017) augmentation method, that was noted to improve the generalization of models on various datasets considerably. Cutout uses randomly positioned zero-valued squares within images, thus increasing the number of possible transforms by $\times 30^2$.

We first tested batch augmentation on the task discussed by Hoffer et al. (2017) – using a ResNet44 (He et al., 2016) over the Cifar10 dataset (Krizhevsky, 2009) together with cutout augmentation (DeVries & Taylor, 2017). We used the original regime by He et al. (2016) with a batch of $B = 64$. We then compared the learning curve with training using batch augmentation with $M \in \{2, 4, 8, 16, 32\}$ different transforms for each sample in the batch, effectively creating a batch of $64 \cdot M$.

As we can see in figure 1, validation convergence speed has noticeably improved (in terms of epochs), with a significant reduction in final validation classification error (figure 1b). This trend largely continues to improve as $M$ is increased, consistent with our expectation.

We verified these results using a variety of models (table 1) using various values of $M$, depending on our ability to fit the $M \cdot B$ within our compute budget (specifically, GPU memory). Our best result was achieved using DARTS final Cifar10 model Liu et al. (2018). DARTS is a differentiable architecture search framework which constructs a graph with a SoftMax parameterized edges. The final model is a subset of the graph whose edges have the highest values.

In all our experiments we have observed significant improvements to the final validation accuracy as well, as an increase in accuracy per epoch convergence speed. A typical example can be seen in

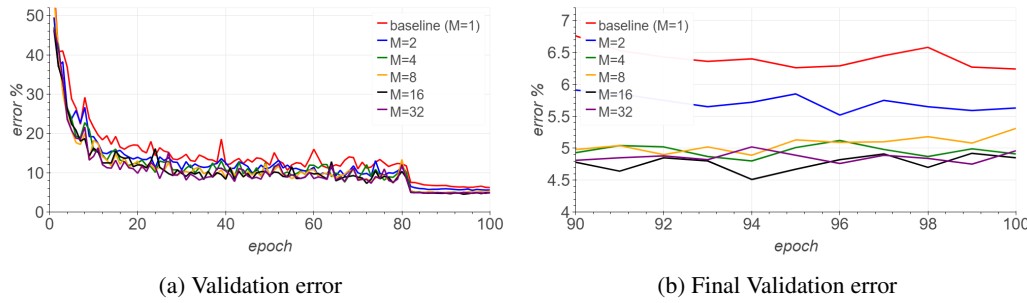

(a) Validation error                 (b) Final Validation error

Figure 1: Impact of batch augmentation (ResNet44 + cutout, Cifar10). We used the original (red) training regime with $B = 64$, and compared to batch augmentation with $M \in \{2, 4, 8, 16, 32\}$ creating an effective batch of $64 \cdot M$

figure 2, where using a mere $M = 6$, BA improved the results of a Wide-ResNet model by more than $0.5\%$ (a relative $17\%$ decrease in error).

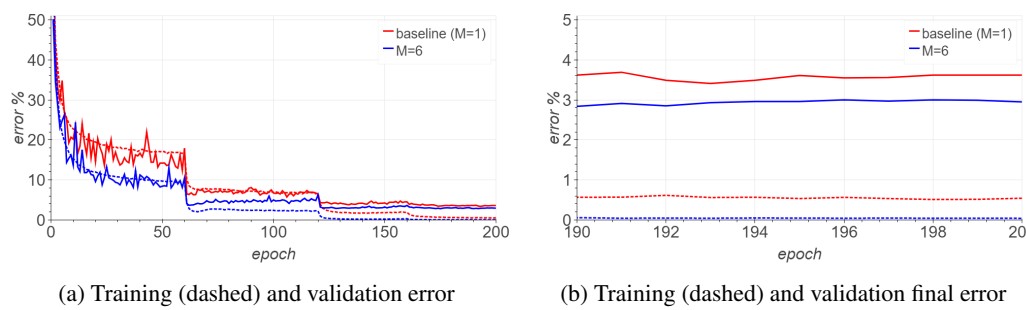

(a) Training (dashed) and validation error        (b) Training (dashed) and validation final error

Figure 2: Impact of batch augmentation (Wide-Resnet28-10 + cutout, Cifar10). We used the original (red) training regime and compared to batch augmentation with $M = 6$

Moreover, we managed to achieve high validation accuracy much quicker with batch augmentation. We trained a ResNet44 on Cifar10 for half of the iterations needed for the baseline using batch augmentation and larger learning rate. we managed to achieve $93.65\%$ accuracy while our baseline achieved $93.07\%$ with double the number of iterations. When the baseline is trained with the same shorten regime there is a significant accuracy degradation. This indicates not only the obvious accuracy gain but a potentially time performance improvement for a given hardware.

Finally, we compare with regime adaptation (RA) method by Hoffer et al. (2017). In this method, the number of epochs is increased so that the number of iteration is fixed when using a larger batch. This makes both RA and BA methods comparable with respect to the number of instances seen for each sample over the course of training. Using the same settings (ResNet44, Cifar10), we find an accuracy gain of $0.6\%$ over the $93.07\%$ result reported by Hoffer et al. (2017). Figures 5 and 6 in Appendix depict these results.

## 3.2 IMAGENET

As a larger scale evaluation, we used the ImageNet dataset (Deng et al., 2009), holding more than 1.2 million images depicting 1000 different categories.

For ResNet50 (He et al., 2016), we used the data augmentation method advocated by Szegedy et al. (2015) that employed various sized patches of the image with size distributed evenly between $8\%$ and $100\%$ and aspect ratio constrained to the interval $[3/4, 4/3]$. The images were also flipped horizontally with $p = 0.5$, and no additional color jitter was performed. For the MobileNet model (Howard et al., 2017), we used a less aggressive augmentation method, as described in the original paper. For Alexnet model (Krizhevsky et al., 2012), we used the original augmentation regime.

For all ImageNet models, we followed the training regime by Goyal et al. (2017) in which an initial learning rate of $0.1$ is decreased by a factor of $10$ in epochs $30, 60, 80$ for a total of $90$ epochs. Weight decay factor of $10^{-4}$ is applied to every parameter in the network except for those of batch-norm layers.

To fit within our time and compute budget constraints, we used a mild $M = 4$ batch augmentation factor for ResNet and MobileNet, and $M = 8$ for AlexNet. Due to memory constraints, the ResNet50 model was trained using multiple feed-forwards and gradient accumulations, creating a "Ghost batch norm" (Hoffer et al., 2017) effect. We again observe an improvement with all models in their final validation accuracy (table 1).

The AlexNet model had the most dramatic improvement – yielding more than $4\%$ improvement in validation accuracy compared to our baseline, and more than $2\%$ than previously best published results (You et al., 2017).

We also highlight the fact that models reached a high validation accuracy quicker. For example, the ResNet50 model reached a $75.7\%$ at epoch $35$ – only $0.6\%$ shy of the final accuracy achieved at epoch $90$ with the baseline model (figure 3). Also, the increase in validation error between epochs $30 - 60$ suggests that either learning rate or weight-decay values may need to be altered as discussed by Zagoruyko (2016) who witnessed similar effects. This leads us to believe that with careful hyper-parameter tuning of the training regime, we can shorten the number of epochs needed to reach the desired accuracy and even improve it further.

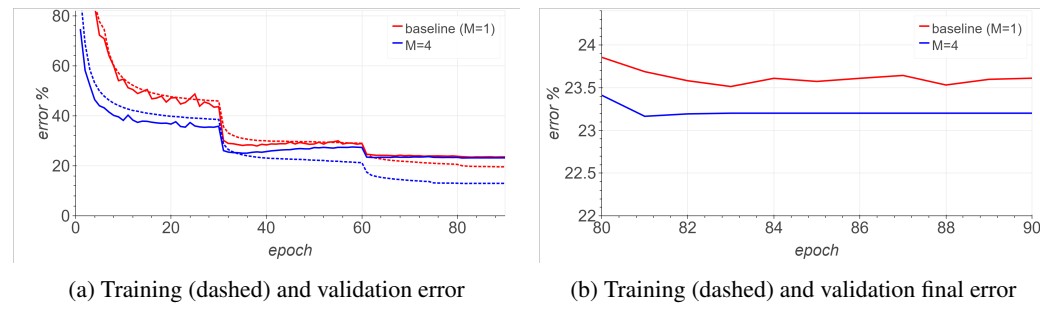

(a) Training (dashed) and validation error

(b) Training (dashed) and validation final error

Figure 3: Impact of batch augmentation (ResNet50, ImageNet). We used the original (red) training regime and compared to batch augmentation with $M = 4$

Table 1: Validation accuracy (Top1) results for Cifar, ImageNet models. Bottom: test perplexity result on Penn-Tree-Bank (PTB) dataset

| Network | Dataset | Baseline | BatchAugment |
|---|---|---|---|
| ResNet44 (He et al., 2016) | Cifar10 | 93.07% | 93.65% (M=10) |
| ResNet44 + cutout | Cifar10 | 93.7% | 95.27% (M=40) |
| VGG + cutout (Simonyan & Zisserman, 2014) | Cifar10 | 93.82% | 95.32% (M=32) |
| Wide-ResNet28-10 + cutout (Zagoruyko, 2016) | Cifar10 | 96.6% | 97.15% (M=6) |
| DARTS (Liu et al., 2018) | Cifar10 | 97.11% | 97.64% (M=10) |
| ResNet44 + cutout | Cifar100 | 72.97% | 74.13% (M=40) |
| VGG + cutout | Cifar100 | 73.03% | 75.5% (M=32) |
| Wide-ResNet28-10 + cutout | Cifar100 | 79.85% | 80.13% (M=10) |
| DenseNet100-12 (Huang et al.) | Cifar100 | 77.73% | 78.8% (M=4) |
| AlexNet (Krizhevsky et al., 2012) | ImageNet | 58.25% | 62.31% (M=8) |
| MobileNet (Howard et al., 2017) | ImageNet | 70.6% | 71.4% (M=4) |
| ResNet50 (He et al., 2016) | ImageNet | 76.3% | 76.8% (M=4) |
| Word-level LSTM (Merity et al., 2017) | PTB | 58.8 ppl | 58.6 ppl (M=10) |

### 3.3 Dropout as intermediate augmentation

We also wished to test the ability of batch augmentation to improve results in tasks where no explicate augmentations are performed on input data. An example for this kind of tasks is language modeling, where the input is fed in a deterministic fashion and noise is introduced in intermediate layers in the form of Dropout (Srivastava et al., 2014), DropConnect (Wan et al., 2013), or other forms of regularization (Krueger et al., 2016; Merity et al., 2017).

We used the implementation by Merity et al. (2017) and the proposed setting of LSTM word-level language model over the Penn-Tree-Bank (PTB) dataset. We used a 3-layered LSTM of width 1150 and embedding size of 400, together with Dropout regularization on both input ($p = 0.4$) and hidden state ($p = 0.25$), with no finetuning.

We used $M = 10$, increasing the effective batch-size from 20 to 200. The use of multiple instances of the same samples within the batch caused each instance to be computed with a different random dropout mask.

We again observed an improvement, yet more modest compared to the previous experiments, reaching a 0.2 improvement compared to baseline in final test perplexity (see table 1).

## 4 Understanding Batch Augmentation

### 4.1 The perils of large batch training

To understand why Batch Augmentation (BA) works, we first aim to better understand why it is harder to train with a large batch: it was previously observed that with large batch there is a need to adjust the learning rate (Hoffer et al., 2017; Goyal et al., 2017), and also that the generalization performance may be degraded when very large batch sizes are used (Goyal et al., 2017). Then, in the next section, we suggest why, with BA, such issues are alleviated. Thus, with BA we can observe more data augmentations during training (which is beneficial for generalization), without suffering much from the typical issues related to large batch training.

We examine the optimization of loss functions of the form

$$f\left(\mathbf{w}\right) = \frac{1}{N} \sum_{n=1}^{N} \ell\left(\mathbf{w}, \mathbf{x}_n, \mathbf{y}_n\right) \tag{1}$$

where $\{\mathbf{x}_n, \mathbf{y}_n\}_{n=1}^{N}$ is a dataset of $N$ data sample-target pairs and $\ell$ is the loss function, of eq. 1 using SGD with mini-batch of size $B$

$$\mathbf{w}_{t+1} = \mathbf{w}_t - \eta \frac{1}{B} \sum_{n \in \mathcal{B}(k(t))} \nabla_{\mathbf{w}} \ell\left(\mathbf{w}_t, \mathbf{x}_n, \mathbf{y}_n\right), \tag{2}$$

where we assume for simplicity that the indices are sampled with replacement, and that $B$ divides $N$. Therefore, $k\left(t\right)$ is sampled uniformly from $\{1, \ldots, N/B\}$. When our model is sufficiently rich and over-parameterized (e.g., deep networks), we typically converge to a minimum $\mathbf{w}^*$ which is global minimum on all datapoints, i.e., $\forall n : \nabla_{\mathbf{w}} \ell\left(\mathbf{w}^*, \mathbf{x}_n, \mathbf{y}_n\right) = 0$. We linearize the dynamics of eq. 2 near $\mathbf{w}^*$ to obtain

$$\mathbf{w}_{t+1} = \mathbf{w}_t - \eta \frac{1}{B} \sum_{n \in \mathcal{B}(k(t))} \mathbf{H}_n \mathbf{w}_t, \tag{3}$$

where we assumed (without loss of generality) that $\mathbf{w}^* = 0$, and denoted $\mathbf{H}_n \triangleq \nabla_{\mathbf{w}}^2 \ell\left(\mathbf{w}, \mathbf{x}_n, \mathbf{y}_n\right)$. Since we are at a global minimum, $\mathbf{H}_n$ are all symmetric PSDs (there are no descent directions). We prove (in the appendix) the following theorem:

**Theorem 1.** *Let*

$$\langle \mathbf{H} \rangle_k \triangleq \frac{1}{|B|} \sum_{n \in \mathcal{B}(k)} \mathbf{H}_n$$

*be the averaged Hessian over the mini-batch and*

$$\lambda_{\max} = \max_{k \in [N/B]} \max_{\forall \mathbf{v}: \|\mathbf{v}\|=1} \mathbf{v}^\top \langle \mathbf{H} \rangle_k \mathbf{v} \tag{4}$$

*be the maximum over the maximal eigenvalues of $\{\langle\mathbf{H}\rangle_k\}_{k=1}^{N/B}$.*

*The iterates of SGD (eq. 3) will converge if*

$$\lambda_{\max} < \frac{2}{\eta}.\tag{5}$$

*Also, this bound is tight in the sense that it is also a necessary condition for certain datasets.*

According to Theorem 1, for a given minimum $\mathbf{w}^*$, the maximum eigenvalue over all the mini-batches (i.e., $\lambda_{\max}$) dictates what is the maximal learning rate for which SGD converges to that minimum. At high learning rates, this encourages SGD to converge towards "flat" minima, where the max eigenvalue of the full Hessian matrix is small, as observed earlier (Keskar et al., 2017), and explained for full batch gradient descent (Nar & Sastry, 2018).

More importantly, eq. 5 also encourages SGD with high learning rate to converge to minima with low variability of $\langle\mathbf{H}\rangle_k$ — since high variability of $\langle\mathbf{H}\rangle_k$ will typically result in large $\lambda_{\max}$. By increasing the mini-batch size, we typically decrease the variability of $\langle\mathbf{H}\rangle_k$, and therefore $\lambda_{\max}$, since we replace max operations with averaging. Thus, if we increase the mini-batch size, certain minima with high variability in $\mathbf{H}_n$ will become more stable, and therefore SGD may converge to them, instead of the original minima with low variability. Since we converge to different minima, these may have worse generalization performance then the original minima.

This issue can be partially mitigated by increasing the learning rate (as suggested by Hoffer et al. (2017); Goyal et al. (2017)) since by sufficiently fine tuning of the learning rate we might make these new minima unstable again, while keeping the original minima stable. However, merely changing the learning rate may not be sufficient for very large batch sizes, when some minima with high variability and low variability will eventually have similar $\lambda_{\max}$, so SGD will not be able to discriminate between these minima.

### 4.2 VARIANCE REDUCTION IN BATCH AUGMENTATION

Using larger mini-batches, both in standard practice, and in batch augmentation, results in a smaller variance of the mini-batch averaged version of the gradients and Hessians ($\langle\mathbf{H}\rangle_k$). Therefore, in both cases $\lambda_{\max}$ decreases, in a way that may result in the large-batch issues described above — the need to tune the learning rate, and the degraded performance with very large batch sizes.

However, standard mini-batch SGD averages the gradient over different samples, while Batch augmentation additionally averages the gradient over several transformed instances of the same samples. These instances, as they describe the same samples (typically with only small changes), may produce correlated gradients within the batch. As such, the variance reduction achieved by batch augmentation with a factor of $M$ is expected to be significantly lower than the ($1/\sqrt{M}$) reduction that an uncorrelated sum of $M$ samples would have. This implies that the $\lambda_{\max}$ (eq. 5) would change much less in batch augmentation then in standard large batch training. Thus, batch augmentation enables the model to see more augmentations while changing $\lambda_{\max}$ much less. Therefore, we are expected to observe less of the issues of standard large batch training.

Nevertheless, batch augmentation still leads to significant variance reduction. In order to empirically evaluate this effect, we measured the $L^2$ of the gradients of the weights throughout the training for the setting described in 3.1. As could be expected, the variance reduction is reflected in the norm values as can be seen in figure 4. As the effective learning step is affected by this variance reduction, we can adapt the learning rate to partially account for this change, as described in the previous section. The correlated nature of the batch suggests that the needed learning rate correction for batch augmentation should be small.

## 5 CONCLUSIONS

In this work, we have introduced "Batch augmentation", a simple and effective method to improve generalization performance of deep networks by training with large batches composed of multiple transforms of each sample. We have demonstrated significant improvements on various datasets and models, with both faster convergence per epoch, as well as better final validation accuracy.

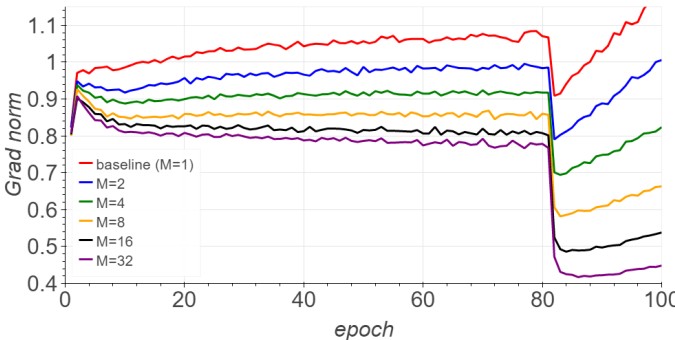

Figure 4: Comparison of gradient $L^2$ norm (ResNet44 + cutout, Cifar10, $B = 64$) between the baseline ($M = 1$) and batch augmentation with $M \in \{2, 4, 8, 16, 32\}$

We suggest a theoretical analysis to explain the advantage of BA over traditional large batch methods. We also show that batch augmentation causes a decrease in gradient variance throughout the training, which is then reflected in the gradients' $L^2$ norm used in each optimization step. This may be used in the future to search and adapt more suitable training hyper-parameters, that will possibly allow faster convergence and even better performance.

Recent hardware development allowed the community to use larger batches without increasing the wall clock time either by using data parallelism or by leveraging more advanced hardware. However, several papers claimed that working with large batch results with accuracy degradation (Masters & Luschi, 2018). Here we argue that by using multiple instances of the same sample we can leverage the larger batch capability to increase accuracy. These findings give another reason to prefer training settings utilizing significantly larger batches than those advocated in the past.

As hardware continues to improve, we are bound to witness more cases similar to the one depicted in table 2 (taken from You et al. (2017)). In this example, we can see that training AlexNet on current hardware takes approximately the same time when using a batch-size of $512$ or $4096$. Our work suggests a novel approach to address this growing issue: simply increase number of augmentation (in this case, to $M = 8$) and achieve better accuracy at the same training time (table 1).

Table 2: The speed and time for AlexNet-BN. Table from You et al. (2017)

| Batch Size | Stable Accuracy | 4-GPU speed | 4-GPU time |
|---|---|---|---|
| 512 | 0.602 | 6627 img/sec | 5h 22m 30s |
| 4096 | 0.604 | 6585 img/sec | 5h 24m 44s |

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

# Appendix

## A    PROOF OF THEOREM 1

By examining the first moment dynamics of this equation

$$\mathbb{E}\mathbf{w}_{t+1} = (\mathbf{I} - \eta \langle \mathbf{H} \rangle) \mathbb{E}\mathbf{w}_t \,, \tag{6}$$

where

$$\langle \mathbf{H} \rangle \triangleq \frac{1}{N} \sum_{n=1}^{N} \mathbf{H}_n$$

it is easy to see that a necessary and sufficient condition for convergence of eq. 6

$$\bar{\lambda}_{\max} < \frac{2}{\eta} \,, \tag{7}$$

where $\bar{\lambda}_{\max}$ is the maximal eigenvalue of $\langle \mathbf{H} \rangle$. This is the standard convergence condition for full batch SGD, i.e., gradient descent.

First, to see eq. 5 is a necessary condition for certain datasets, suppose we have $\mathbf{H}_n = 0$ in all samples, except, in a single mini-batch $k$, for which we have

$$\lambda_{\max} = \max_{\forall \mathbf{v}: \|\mathbf{v}\|=1} \mathbf{v}^\top \langle \mathbf{H} \rangle_k \mathbf{v} \,,$$

In this case, the weights are updated only when we are at mini-batch $k$. Therefore, ignoring all the mini-batches, the dynamics are equivalent to full batch gradient descent with the dataset restricted to mini-batch $k$. Therefore, $\bar{\lambda}_{\max} = \lambda_{\max}$, and we only have first order dynamics (with no noise). Thus, the necessary and sufficient condition for stability is eq. 7 with $\bar{\lambda}_{\max} = \lambda_{\max}$, which is eq. 5.

Next, to show eq. 5 is also a sufficient condition (for all data sets) we examine the second moment dynamics. First we observe that

$$\mathbf{w}_{t+1}^\top \mathbf{w}_{t+1} = \mathbf{w}_t^\top \left( \mathbf{I} - \eta \langle \mathbf{H} \rangle_{k(t)} \right)^\top \left( \mathbf{I} - \eta \langle \mathbf{H} \rangle_{k(t)} \right) \mathbf{w}_t \,.$$

$$= \mathbf{w}_t^\top \left( \mathbf{I} - 2\eta \langle \mathbf{H} \rangle_{k(t)} + \eta^2 \langle \mathbf{H} \rangle_{k(t)} \langle \mathbf{H} \rangle_{k(t)} \right) \mathbf{w}_t \,.$$

Denoting

$$\langle \mathbf{H}^2 \rangle \triangleq \frac{1}{N/B} \sum_{k=0}^{N/B} \langle \mathbf{H} \rangle_k \langle \mathbf{H} \rangle_k \,.$$

Thus, we obtain

$$\mathbb{E} \|\mathbf{w}_{t+1}\|^2 = \mathbb{E} \left[ \mathbf{w}_{t+1}^\top \left( \mathbf{I} - 2\eta \langle \mathbf{H} \rangle + \eta^2 \langle \mathbf{H}^2 \rangle \right) \mathbf{w}_t \right] \,. \tag{8}$$

Since $\mathbf{H}_n$ are all PSDs it is easy to see that if $\mathbf{z}$ is a zero eigenvector of $\langle \mathbf{H} \rangle$ or $\langle \mathbf{H}^2 \rangle$ then it must be a zero vector eigenvector of other matrix, and also of all $\mathbf{H}_n$, $\forall n$. We denote the null space

$$\mathcal{V} \triangleq \left\{ \mathbf{v} \in \mathbb{R}^d | \|\mathbf{v}\| = 1, \langle \mathbf{H} \rangle \mathbf{z} = 0 \right\}$$

and its complement $\bar{\mathcal{V}}$. From eq. 8 a necessary and sufficient condition for convergence of this equation is

$$\max_{\mathbf{v} \in \bar{\mathcal{V}}} \mathbf{v}^\top \left( \mathbf{I} - 2\eta \langle \mathbf{H} \rangle + \eta^2 \langle \mathbf{H}^2 \rangle \right) \mathbf{v} < 1 \,. \tag{9}$$

To complete the proof we will show that eq. 5 also implies eq. 9, for any $B$.

First we notice that Eq. 4 implies that $\forall \mathbf{v} \in \bar{\mathcal{V}}$ :

$$\mathbf{v}^\top \langle \mathbf{H}^2 \rangle \mathbf{v} = \frac{1}{N} \sum_{k=0}^{N/B} \sum_{n \in \mathcal{B}(k)} \mathbf{v}^\top \langle \mathbf{H} \rangle_k \mathbf{H}_m \mathbf{v} < \frac{1}{N} \sum_{n=1}^{N} \lambda_{\max} \mathbf{v}^\top \mathbf{H}_n \mathbf{v} = \lambda_{\max} \mathbf{v}^\top \langle \mathbf{H} \rangle \mathbf{v} \,. \tag{10}$$

Also, since $\lambda_{\max} > \bar{\lambda}_{\max}$, we have

$$\mathbf{v}^\top \langle \mathbf{H} \rangle^2 \mathbf{v} \le \lambda_{\max} \mathbf{v}^\top \langle \mathbf{H} \rangle \mathbf{v} . \tag{11}$$

We combine the above results to prove the Lemma, and $\forall \mathbf{v} \in \bar{\mathcal{V}}$ :

$$\mathbf{v}^\top \left[ (\mathbf{I} - 2\eta \langle \mathbf{H} \rangle) + \eta^2 \langle \mathbf{H}^2 \rangle \right] \mathbf{v}$$
$$= 1 - 2\eta \mathbf{v}^\top \langle \mathbf{H} \rangle \mathbf{v} + \eta^2 \mathbf{v}^\top \langle \mathbf{H}^2 \rangle \mathbf{v}$$
$$\overset{(1)}{\le} 1 - 2\eta \mathbf{v}^\top \langle \mathbf{H} \rangle \mathbf{v} + \eta^2 \lambda_{\max} \mathbf{v}^\top \langle \mathbf{H} \rangle \mathbf{v}$$
$$= 1 - \eta \left( 2 - \eta \lambda_{\max} \right) \mathbf{v}^\top \langle \mathbf{H} \rangle \mathbf{v} ,$$

where in (1) we used eqs. 10 and 11. Given the condition in eq. 5 this is smaller then 1, so eq. 9 holds, so this proves the Theorem.

As a side note, we can bound the convergence rate using the last equation. To see this, we denote $\mathcal{P}_{\bar{\mathcal{V}}}$ as the projection to $\bar{\mathcal{V}}$, and

$$\lambda_{\min} \triangleq \min_{\forall \mathbf{v} \in \mathcal{V}} \mathbf{v}^\top \langle \mathbf{H} \rangle \mathbf{v}$$

as the smallest non-zero eigenvalue of $\langle \mathbf{H} \rangle$. iterating the recursion we obtain that the convergence rate is linear

$$\mathbb{E} \left\| \mathcal{P}_{\bar{\mathcal{V}}} \mathbf{w}_t \right\|^2 \le \left( 1 - \eta \left( 2 - \eta \lambda_{\max} \right) \lambda_{\min} \right)^t \mathbb{E} \left\| \mathcal{P}_{\bar{\mathcal{V}}} \mathbf{w}_0 \right\|^2 . \tag{12}$$

However, note this bound is not necessarily tight.

### A.1 Comparison with longer training

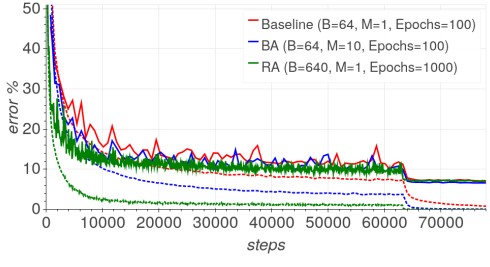
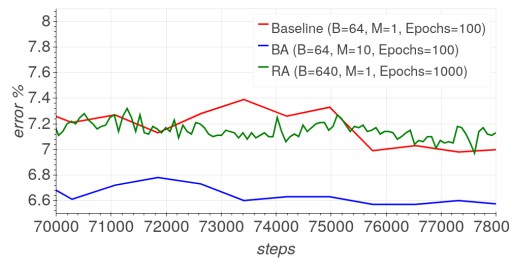

(a) Training (dashed) and validation error

(b) Training (dashed) and validation final error

Figure 5: A comparison between (1) baseline B=64 training (2) our batch augmentation (BA) method with M=10 (3) regime adaptation (RA) with B=640 and 10x more epochs

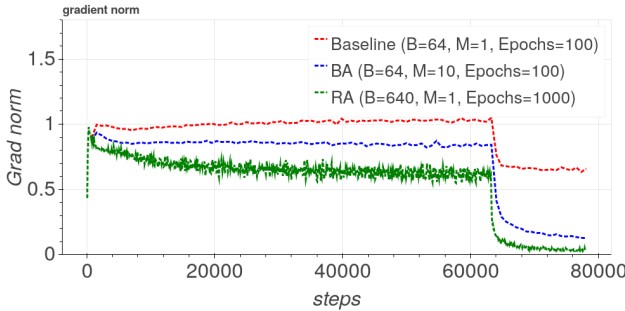

Figure 6: A comparison of gradient norm between (1) baseline B=64 training (2) our batch augmentation (BA) method with M=10 (3) regime adaptation (RA) with B=640 and 10x more epochs. As expected, BA exhibits a gradient norm smaller than Baseline, but larger than large-batch training.

