# OpenReview forum: "Augment your batch: better training with larger batches"
_ICLR.cc/2019/Conference_

### Official Review · AnonReviewer2 · 2018-10-31

**Rating:** 4
**Confidence:** 4

**Review:**

The paper shows that training with large batch size (e.g., with MxB samples) serves as an effective regularization method for deep networks, thus improving the convergence and generalization accuracy of the models. The enlarged batch of MxB consists of multiple (i.e., B) transforms of each of the M samples from the given batch; the transform is executed by a data augmentation method such as Cutout or Dropout. The authors also provide a theoretical explanation for the working of the method, suggesting that the enlarged batch training decreases the gradient variance during the training of the networks.

The paper is well written and easy to follow. Also, some interesting results are experimentally obtained such as the figures presented in Figure4. Nevertheless, the experimental studies are not very satisfactory in its current form.

Major remarks:

1.	In terms of regularization with transformed data in a given batch, the proposed method is related to MixUp (Zhang et al., mixup: Beyond empirical risk minimization), AdaMixUp (Guo et al., MixUp as Locally Linear Out-Of-Manifold Regularization), Manifold Mixup (Verma et al., Manifold Mixup: Learning Better Representations by Interpolating Hidden States), and AgrLearn (Guo et al. Aggregated Learning: A Vector Quantization Approach to Learning with Neural Networks). It would be useful for the authors to discuss how the proposed strategy differs from them or empirically show how the proposed regularization method compares to them in terms of regularization effect.  For example, in MixUp, AdaMixup and Manifold Mixup, the samples in a given batch will be linearly interpolated with randomly reshuffled samples of the same batch. In these sense, using them as baselines would make the contribution of the proposed method much significant.
2.	In the experiments, it seems the authors use different data augmentation methods for different datasets  (except for Cifar10 and Cifar100), it would be useful to stick with a particular data augmentation method for all the datasets, for example, it would be interesting to see the performance of also using Cutout for the MobileNet and ResNet50 on the ImageNet data set.
3.	Regarding the experimental study, I wonder if it would be beneficial to include three variations of the proposed method. First, use baseline with the same batch size, namely BxM, but with sampling with replacement. That is, using the same batchsize as that in Batch Augmentation but with repeated samples. In this way, the contribution of the data augmentation in the proposed method would be much clearer. Second, as suggested from the results in the PTB data in Table1, using only Dropout obtains very minor improvement over the baseline method. In this sense, using other data augmentation methods instead of Cutout for the image tasks would make the contribution of the paper much clear. Third, training the networks with the batchsize of BxM, but excluding the original data samples in the given batch would be another interesting experiment. That is, all samples of the batch in the batch augmentation are synthetic samples.

Minor remarks:

1.	Is the regularized model robust to adversarial attacks as suggested in Mixup and Manifold Mixup?
2.	Would it be beneficial to include various data augmentation methods for the same batch? That is, each transformed sample may come from a different data augmentation strategy.

==========after rebuttal===========

My main concern is that the paper did not clearly show where the performance improvement comes from. It may simply come from the larger batch size instead of the added augmented samples as claimed by the paper. I think the current comparison baseline in the paper is insufficient. I did propose three comparison baselines in my initial review, but I am not satisfied with the authors' rebuttal on that.

---

> ### Public Comment · ~Alex_Matthew_Lamb1 · 2018-11-06
> **Note on Manifold Mixup and Adv. Robustness**
>
> "1.	Is the regularized model robust to adversarial attacks as suggested in Mixup and Manifold Mixup?"
>
> Both mixup and manifold mixup have improved robustness to the single-step FGSM attack.  Neither leads to robustness on multi-step attacks like PGD.

---

> ### Author Response · Authors · 2018-11-26
> **Reply to AnonReviewer2**
>
> 1. Regarding mixup: mixup requires a mixed input from two separate labels as well as a mixed target by same amount. It does not deal with data augmentations as BA (multiple instances of same sample). Therefore, Mixup approach is orthogonal to ours and both can be combined.
>
> 2. We used different augmentation techniques to emphasize the improvement of batch-augment upon them all. We commonly used augmentation for each network (for example, cutout is common for modern cifar-10 based models, but not for ImageNet). We agree that applying BA with other augmentation techniques would make an interesting experiments, that can further improve accuracy, but we argue that this is not the essence of our work.
>
> 3. We've added an additional experiment regarding training for longer with M*B batch size (accounting for same number of examples) in Appendix Figure 5. We wish to clarify that in each experiment we've performed (including baseline) the same augmentation technique was used (according to original paper, or explicitly stated as in the case of cutout).
>
> 4. We stress that in this work, we do not suggest a new type of augmentation technique but rather a method that utilize any type of augmentation. Thus,  we argue that BA should be as robust to adversarial attacks as the augmentation technique it utilize (e.g., cutout, random cropping, flipping, etc.).  Nonetheless, we thank the reviewer for his suggestions and encourage researchers to use BA with different augmentation strategies.

---

### Official Review · AnonReviewer1 · 2018-11-02
**simple idea that works along with some theory to support it**

**Rating:** 8
**Confidence:** 3

**Review:**

This paper describes a new method for data augmentation which is called batch augmentation. The idea is very simple -- include in your batch M augmentations of the each training sample, effectively this will increase the size of the batch by M. I have not seen a similar idea to this proposed before. As the authors show this simple technique has the potential to increase training convergence and final accuracy. Several experiments support the paper's claims illustrating the effectiveness of the technique on a variety of datasets (e.g. CIFAR, ImageNet, PTB) and architectures (ResNet, Wide-ResNet, DenseNet, MobileNets). Following that there's a more theoretical section which provides some analysis on why the method works, and seems also reasonable. Overall simple idea, well written-paper with clear practical application and of potential great interest to many researchers

---

> ### Author Response · Authors · 2018-11-26
> **Reply to AnonReviewer1**
>
> We thank the reviewer for his remarks and positive assessment of our work.

---

### Official Review · AnonReviewer3 · 2018-11-02
**Interesting idea with insufficient support**

**Rating:** 4
**Confidence:** 4

**Review:**

This paper tested a very simple idea: when we do large batch training, instead of sampling more training data for each minibatch, we use data augmentation techniques to generate training data from a small minibatch. The authors claim the proposed method has better generalization performance.

I think it is an interesting idea, but the current draft does not provide sufficient support.

1. The proposed method is very simple. In this case, I would expect the authors provide more intuitive explanations. It looks to me the better generalization comes from more complicated data augmentation, not from the proposed large batch training.

2. It is unclear to me what is the benefit of the proposed method. Even provided more computing resources, the proposed method is not faster than small batch training. The improvement on test errors does not look significant. If given more computing resources, and under same timing constraint, we have many other methods to improve performance. For example, a simple thing to do is t0 separately train networks with standard setting and then ensemble trained networks. Or apply distributed knowledge distillation like in (Anil 2018
Large scale distributed neural network training through online distillation)

3. The experiments are not strong. The largest batch considered is 64*32, which is relatively small. In figure 1 (b), the results of M=4,8,16,32 are very similar, and it looks unstable. It is unclear what is the default batchsize for Imagenet. In Table 1, the proposed method tuned M as a hyperparameter. The baselines are fairly weak, the authors did not compare with any other method. I would expect at least the following baselines:
i)  use normal large batch training and complicated data augmentation, train the model for same number of epochs
ii) use normal large batch training and complicated data augmentation, train the model for same number of iterations
ii) use normal large batch training and complicated data augmentation, scale the learning rate up as in Goyal et al. 2017

4. For theorem 1, it is hard to say how much the theoretical analysis based on linear approximation near global minimizer would help understand the behavior of SGD. I fail to understand the the authors’ augmentation. Following the author’s logic, normal large batch training decrease the variability of <H>_k and \lambda_max, which converges to ‘’flat’’ minima. It contradicts with the authors’ other explanation.

5. In section 4.2, I fail to understand why the proposed method can affect the norm of gradient.


6. Related works:
Smith et al. 2018 Don't Decay the Learning Rate, Increase the Batch Size.


=============== after rebuttal ====================
I appreciate the authors' response, but I do not think the rebuttal addressed my concerns. I will keep my score and argue for the rejection of this paper.

My main concern is that the benefit of this method is unclear. The main baseline  that has been compared is the standard small-batch training. However, the proposed method use a N times larger batch and same number of iterations, and hence N times more computation resources. Moreover, the proposed method also use N times more augmented samples. Like the authors said, they did not propose new data augmentation method, and their contribution is how to combine data augmentation with large-batch training. However, I am not convinced by the experiments that the good performance is from the proposed method, not from the N times more augmented samples. I have suggested the authors to compare with stronger baselines to demonstrate the benefits. However, the authors quote a previous paper that use different data augmentation and (potentially) other experimental settings.

The proposed method looks unstable. Moreover, instead of showing the consistent benefits of large batch, the authors tune the batchsize as a hyperparameter for different experiments.

Regarding the theoretical part, I still do not follow the authors' explanation. I think it could at least be improved for clarity.

---

> ### Author Response · Authors · 2018-11-26
> **Reply to AnonReviewer3**
>
> 1. There may have been a misunderstanding: we compared our method to the baseline, and both had the same type of data augmentation (e.g. in ResNet we did this comparison for both normal augmentation and cutout separately). We are therefore certain that the generalization improvements stem from batch-augment method, as it appears for all augmentation schemes we've tried.
>
> 2. Both ensemble methods and "Distributed knowledge distillation" result in models larger then the original model, and therefore requires additional resources at run time (after training). In contrast, our method does not have this issue as the final trained method is identical to the original. Moreover, our method is much simpler and does not require any change of settings - as we used the original training regime without modifications. Lastly, in many cases it is possible to increase the batch size without affecting the wall clock time, due to surplus compute power (e.g., Table 2). In those cases, our BA method can be easily used to take advantage of this surplus large batch size, and improve the final model accuracy, as we demonstrated.
>
> 3.We kindly disagree, as we feel that results on various models and datasets show a consistent and (mostly) non-trivial improvement on baseline results. As others have shown before, a batch size of 64*32=2048 is not small, and often yields noticeably decreased in accuracy when training regime is not adapted [1].
> We note that similar experiments to the ones the reviewer asked for were previously done in [1, Table 1&2] for several datasets and models. For example, for a baseline of 93.07% (Resnet44 on cifar10 dataset), we improved to 93.65%. However:
> (i) On large batch without adapting the training regime for the same number of steps: accuracy drops to 86.10% [1].
> (ii+iii) When using large batch for the same number of steps and learning rate is increased, accuracy returns to 93.07%. For experiment (ii) we note that accuracy is marginally worse. We've added this experiment to the paper along with convergence graphs (Figure 5, Appendix).
>
> 4. "For theorem 1, it is hard to say how much the theoretical analysis based on linear approximation near global minimizer would help understand the behavior of SGD."
> Our theoretical analysis is focused on how SGD selects stationary points, using stability analysis. Such stability analysis requires linearization.
> "I fail to understand the the authors’ augmentation. Following the author’s logic, normal large batch training decrease the variability of <H>_k and \lambda_max, which converges to ‘’flat’’ minima. It contradicts with the authors’ other explanation."
> There may have been a misunderstanding: increasing batch size will not decrease flatness, i.e. the maximal eigenvalue of the Hessian (as defined in Keskar et al.), which is different from \lambda_max. To clarify, we suggested in section 4 that BA works well since enables the model to observe more augmentations, with only a small effect on the variance (since most of the samples in the mini-batch are highly correlated). This is in contrast to standard large-batch training,  which works less well since it has a larger effect on the variance.
>
> 5. As we explained in section 4.2, batch-augmentation causes each batch to have correlated samples (different instances of the same image) . When computing gradients on this batch we accumulate multiple gradient instances -- leading to smaller variance, and hence, smaller norm.
> This reduction is less than the reduction in large-batch training, since the batch instances are much more highly correlated. We've added an additional figure (Figure 6, Appendix) that demonstrates this point.
>
> [1] "Train Longer Generalize Better" - Hoffer et al (NIPS 2017).

---

### Public Comment · ~Eddie_Smolyansky1 · 2018-09-28
**Are the comparisons done with the same amount of image instances?**

Hi, it seems to me that due to the augmentation, there are now M times more image instances per epoch of training, right?
Perhaps I missed it in the paper, but have your comparisons (in Fig1, Fig2 for example)  been done with the same amount of image instances per epoch? A clarification could be to show Fig1, Fig2 with x axis being "image instances" and not "epoches".

My question boils down to: how have you demonstrated that the improved accuracy is due to the augmentations, as opposed to simply having a larger batch size?

---

> ### Author Response · Authors · 2018-09-29
> **answer - comparing same number of instances**
>
> We thank you for the interest and question.
> Training with large batch was noted in previous works to cause degradation in validation accuracy. While previous works focused on reducing the wall clock time without suffering from this degradation ("generalization gap"), our method is first to suggest significant improvement with large batches.
> If we understand you correctly, you are interested to see if our improvements can also be gained with training using larger batches for the same number of iterations (called "regime adaptation" in [1]).
> That way, the same number of image instances is seen by the model as in our method (but with a larger number of epochs). This kind of comparison is described in the last paragraph of section 3.1.
> The full training results are available at https://drive.google.com/file/d/1mcHSnIx_dxjwTeYuUIrJmmcaLKQ-jDU5/view?usp=drivesdk
> here you can see a comparison between
> (1) baseline B=64 training
> (2) our batch augmentation (BA) method with M=10
> (3) regime adaptation (RA) with B=640 and 10x more epochs
>
> In the validation accuracy graph, you can observe that although the same number of sample instances were seen for both (2) and (3), our BA method still achieved a considerable improvement.
> We hope we answered your concerns.
>
> [1] "Train longer generalize better" (2017) - Hoffer, Hubara, Soudry

---

> > ### Public Comment · (anonymous) · 2018-09-29
> > **Comparison**
> >
> > Would you please show the results of "RA (B=64, M=1, Epochs=1000)"?
> > It may be a more proper comparison with BA than "RA (B=640, M=1, Epochs=1000)".

---

> > > ### Author Response · Authors · 2018-10-02
> > > **clarification**
> > >
> > > Thanks for your interest. Please see the clarification we added to address your question.

---

> > ### Public Comment · ~Eddie_Smolyansky1 · 2018-09-30
> > **Followup**
> >
> > Thank you for the response, what a great platform to have a discussion. I believe you have understood my question correctly and addressed it.
> >
> > Small comments:
> > 1. There might be a typo in this paper. In the final paragraph of section 3.1 you report the results of [1] are 93.04 but in the paper itself I find 93.07. Is it a coincidence that it's the same number as the baseline for "ResNet44 (He et al., 2016)" in this paper (shouldn't it be higher)?
> > 2. I'm assuming in (3) above you mean B=64. Also, I can't access the gdrive link but I believe your report.

---

> > > ### Author Response · Authors · 2018-10-02
> > > **answer**
> > >
> > > Thanks.
> > > 1. Indeed, that is a typo. It should be 93.07%, we will fix it in the next revision.
> > > 2. In the graph we posted,  RA was measured with B=640, see the clarification we posted for the case B=64.

---

### Public Comment · (anonymous) · 2018-09-30
**Mixup?**

Haven't had a chance to read the whole paper yet but was surprised to not see any mention of mixup. Seems to me like it would make sense to send 1-2 augmented copies of each sample to the device then expand the size of the batch by using mixup on multiple combinations of those samples.

---

> ### Author Response · Authors · 2018-10-02
> **Mixup**
>
> Thanks for your interest. Notice that mixup requires a mixed input from two separate labels as well as a mixed target by same amount. It does not deal with data augmentations as BA (multiple instances of same sample). Therefore, Mixup approach is orthogonal to ours and both can be combined. We welcome our readers to try and incorporate our ideas in this setting.

---

### Author Response · Authors · 2018-10-02
**clarifications**

We would like to clarify the main points of our paper:
1. In many cases, it is possible to increase the batch size without affecting the wall clock time, due to surplus compute power (e.g., see Table 2). In those cases, our BA method can be used to take advantage of this large batch size, and improve the final model accuracy, as we demonstrated.
2. We suggested in section 4 that the reason that BA works well is that it enables the model to observe more augmentations, with only a small effect on the variance (since most of the samples in the mini-batch are highly correlated). This is in contrast to standard large-batch + augmentation (i.e., Regime Adaptation from [1]) which works less well since it has a larger effect on the variance.

The comments here suggested that instead of using BA with M=10, we simply increase the number of iterations x10 (keeping a small batch size of B=64). This results with the same accuracy gain as doing BA with the same M. This is not surprising since we observe more augmentations, while not changing the mini-batch variance. However, we would not gain from the computational benefits of a larger batch as the training time can be roughly x10 times longer.

We would also take the opportunity to report a result obtained after the paper was submitted:
Using the AlexNet model, with [B=512, M=8], we obtained a top-1 accuracy of 62.308% (up from baseline 57% and from 60% obtained by [2]). This echoes our messages clearly given table 2 in the paper -- using the same wall-clock time you can increase your model accuracy significantly by using BA.

[1] "Train longer generalize better" (2017) - Hoffer, Hubara, Soudry
[2] "Scaling SGD Batch Size to 32K for ImageNet Training" - You, Gitman, Ginsburg

---

### Meta-Review · Area_Chair1 · 2018-12-14
**evaluation should be improved**

**Confidence:** 5
**Recommendation:** Reject

**Metareview:**

The authors propose to use large batch training of neural networks, where each batch contains multiple augmentations of each sample. The experiments demonstrate that this leads to better performance compared to training with small batches. However, as noted by Reviewers 2 and 3, the experiments do not convincingly show where the improvement comes from. Considering that the described technique is very simplistic, having an extensive ablation study and comparison to the strong baselines is essential. The rebuttal didn’t address the reviewers' concerns, and they argue for rejection.